Effects of manipulating slowpoke calcium-dependent potassium channel expression on rhythmic locomotor activity in Drosophila larvae

McKiernan Erin C. emck31@gmail.com
Instituto Tecnológico y de Estudios Superiores de Monterrey , Xochitepec, Morelos , México
Kubke M Fabiana
Electronic publication date: 2013 Mar 26
Publication date: 2013
Volume: 1
Electronic Location ID: e57
Received 2012 Dec 7; Accepted 2013 Mar 5
Copyright: © 2013 McKiernan
Copyright year: 2013
Copyright holder: McKiernan
License: This is an open access article distributed under the terms of the Creative Commons Attribution License, which permits unrestricted use, distribution, and reproduction in any medium, provided the original author and source are credited.
License URL: https://creativecommons.org/licenses/by/3.0/

Keywords: Ion channels, Slowpoke, Calcium-dependent potassium channels, Motor pattern, Locomotion, Intrinsic properties, Motor neurons, Drosophila

Funding: NIH Predoctoral Training Program in Neuroscience 5 T32 AG007437 NIH grant 5 R01 NS057637 03 ECM was funded in part by the NIH Predoctoral Training Program in Neuroscience 5 T32 AG007437 awarded to the Interdisciplinary Graduate Program in Physiological Sciences at the University of Arizona, and by NIH grant 5 R01 NS057637 03 awarded to R.B. Levine at the University of Arizona. All experiments were performed at Arizona State University, School of Life Sciences in the laboratory of C. Duch. The funders had no role in study design, data collection and analysis, decision to publish, or preparation of the manuscript.

==============================
Rhythmic motor behaviors are generated by networks of neurons. The sequence and timing of muscle contractions depends on both synaptic connections between neurons and the neurons’ intrinsic properties. In particular, motor neuron ion currents may contribute significantly to motor output. Large conductance Ca2+-dependent K+ (BK) currents play a role in action potential repolarization, interspike interval, repetitive and burst firing, burst termination and interburst interval in neurons. Mutations in slowpoke (slo) genes encoding BK channels result in motor disturbances. This study examined the effects of manipulating slo channel expression on rhythmic motor activity using Drosophila larva as a model system. Dual intracellular recordings from adjacent body wall muscles were made during spontaneous crawling-related activity in larvae expressing a slo mutation or a slo RNA interference construct. The incidence and duration of rhythmic activity in slo mutants were similar to wild-type control animals, while the timing of the motor pattern was altered. slo mutants showed decreased burst durations, cycle durations, and quiescence intervals, and increased duty cycles, relative to wild-type. Expressing slo RNAi in identified motor neurons phenocopied many of the effects observed in the mutant, including decreases in quiescence interval and cycle duration. Overall, these results show that altering slo expression in the whole larva, and specifically in motor neurons, changes the frequency of crawling activity. These results suggest an important role for motor neuron intrinsic properties in shaping the timing of motor output.

Introduction

Rhythmic motor behaviors, such as respiration and locomotion, are vital to animal survival, and must be reliably and precisely controlled by the nervous system. The sequence and timing of muscle contractions producing these behaviors comprise the motor pattern, and are generated by collections of synaptically connected neurons called central pattern generating (CPG) networks (Grillner, 2006; Harris-Warrick, 2010; Marder & Bucher, 2001). In many systems, motor neurons (MNs) are not part of the classically-defined CPG network (Marder & Bucher, 2001). However, intrinsic MN properties, such as specific ionic currents, may play a crucial role in producing proper motor output (del Negro, Hsiao & Chandler, 1999; Gorassini et al., 1999; Hounsgaard et al., 1984; Wright & Calabrese, 2011) (for reviews see Harris-Warrick (2002), Heckman et al. (2009), Kiehn et al. (2000) and Marder & Goaillard (2006)). To what extent MN currents shape the motor pattern, and exactly how aspects of the pattern are altered by expression of specific ion channel genes, are open questions.

MNs display a variety of K+ currents that shape responsiveness to synaptic inputs and firing output (Harris-Warrick, 2002; McLarnon, 1995). Of particular interest are Ca2+-dependent K+ currents (IKCa) carried through ‘maxi-K’ or ‘Big K’ (BK) channels (Faber & Sah, 2003; Salkoff et al., 2006). BK channels require an increase in cytosolic Ca2+ and membrane depolarization to maximally activate. BK currents have been shown to play a role in action potential repolarization (Benhassine & Berger, 2009; Liu & Shipley, 2008; Shao et al., 1999), fast after-hyperpolarization (Gu, Vervaeke & Storm, 2007; Sausbier et al., 2004; Shao et al., 1999), regulation of firing frequency and interspike interval (Gu, Vervaeke & Storm, 2007; Sun & Dale, 1998; Womack & Khodakhah, 2004), repetitive and burst firing (Benhassine & Berger, 2009; Gu, Vervaeke & Storm, 2007; Wang, Olshausen & Chalupa, 1999), interburst interval (Womack & Khodakhah, 2004), and burst termination (Liu & Shipley, 2008; Sun & Dale, 1998; Womack & Khodakhah, 2004). In addition, mutations in slowpoke (slo) genes encoding BK channels are associated with motor disturbances and disorders (Du et al., 2005; Meredith et al., 2004; Sausbier et al., 2004).

The slo gene was originally cloned in Drosophila (Atkinson, Robertson & Ganetzky, 1991; Elkins, Ganetzky & Wu, 1986; Singh & Wu, 1989). Channels encoded by the slo gene carry transient IKCa in muscles (Elkins, Ganetzky & Wu, 1986; Komatsu et al., 1990; Singh & Wu, 1989), and both transient and sustained IKCa in neurons (Saito & Wu, 1991). slo mutations in Drosophila cause action potential broadening (Atkinson et al., 1998; Brenner et al., 2000; Elkins & Ganetzky, 1988; Elkins, Ganetzky & Wu, 1986; Saito & Wu, 1991), decreased delay to first spike (Elkins & Ganetzky, 1988; Elkins, Ganetzky & Wu, 1986), increased interspike interval (Elkins & Ganetzky, 1988), changes in firing patterns that include “abnormal regenerative responses” (Saito & Wu, 1991; Singh & Wu, 1990), delayed repolarization of the neuromuscular junction (Gho & Ganetzky, 1992), and reduced synaptic transmission (Lee, Ueda & Wu, 2008; Warbington et al., 1996). Locomotor deficits are observed in adult slo mutants, including shaking under ether anesthesia, reduced flight, semi-paralysis in response to heat or bright light (Atkinson et al., 1998; Atkinson et al., 2000; Elkins, Ganetzky & Wu, 1986), and abnormal circadian patterns of activity (Ceriani et al., 2002; Fernández et al., 2007). The role of slo channels in Drosophila MNs and their specific contribution to the timing of larval locomotor activity has not, to the author’s knowledge, been reported.

This study examined the effects of manipulating slo expression on rhythmic locomotor activity in Drosophila larvae. Dual intracellular recordings were made from neighboring body wall muscles during spontaneous fictive crawling. slo expression was manipulated at two levels: (1) in the whole animal with a hypomorphic mutation, or (2) in identified MNs with a RNA interference (RNAi) construct. Overall, the results show that altering slo channel expression, either in the whole animal or in identified MNs, changes the frequency of crawling activity. In particular, these results suggest that MN intrinsic properties may shape the timing of locomotor behavior.

Methods

Fly lines and rearing

Drosophila melanogaster were reared at 25 °C under a 12 h light-dark cycle on standard yeast-sugar-cornmeal media. Wandering third-instar larvae were used for all experiments. w1118 larvae were used as wild-type (WT) control. The slo mutant line, s t1slo1 (hereafter referred to as slo1), was obtained from Bloomington Drosophila Stock Center (Stock No. 4587). The UAS-slo RNAi line was obtained from Vienna Drosophila RNAi Center (Transformant ID: 6723). Expression of RNAi was restricted using the RRA-GAL4 driver (Fujioka et al., 2003) to two identified MNs: (1) the MN innervating muscle 1 with big terminal boutons of Type I (glutamatergic), known as MN1-Ib, and (2) the MN innervating dorsal muscles, including muscle 1, via the intersegmental nerve (ISN) with small Type I terminal boutons, known as MNISN-Is (Hoang & Chiba, 2001). MN1-Ib and MNISN-Is are also known by their embryonic identities as aCC and RP2, respectively (Fujioka et al., 2003). The RRA-GAL4 driver line (obtained from Subhashini Srinivasan) included a Dicer construct (UAS-Dicer/Cyo;RRA-GAL4, UAS-cd8 GFP/RRA-GAL4) to increase the strength of the RNAi (Dietzl et al., 2007).

Larval preparations

Larvae were dissected and recorded in HL3.1 saline (Feng, Ueda & Wu, 2004) containing (in mM): 70 NaCl, 5 KCl, 1.5 CaCl2, 4 MgCl2, 10 NaHCO3, 5 Trehalose, 115 Sucrose, 5 HEPES, pH 7.1-7.3. All chemicals were obtained from Sigma (St. Louis, Missouri). Previous electrophysiological studies of Drosophila larvae have used a dissection method which involves cutting up the dorsal midline (Barclay, Atwood & Robertson, 2002; Cattaert & Birman, 2001; Cooper & Neckameyer, 1999; Fox, Soll & Wu, 2006; Song et al., 2007; Ueda & Wu, 2006). However, since reliable access to intact dorsal-most body wall muscles 1 and 2 (Hoang & Chiba, 2001) was required for experiments, a new dissection method was developed in which these muscles did not risk damage from cutting down the midline or pinning. Larvae were pinned at the head and tail in silicone elastomer (Sylgard)-lined dishes. A cut was made to the right of the dorsal midline, typically through muscle 4 (Hoang & Chiba, 2001), which left all muscles and axons on one side of the larva intact for recording. The opposite side suffered damage to the muscles through which the cut was made. In addition, the peripheral nerves which lie in the muscle field were cut as a result of the incision, leaving many of the muscles in the dorsal group on the cut side without functional innervation. All organs and fat bodies were removed to allow access to the muscles. Larvae dissected with this method generated rhythmic peristaltic waves similar to those recorded from larvae dissected up the midline, except for an acceleration of the rhythm (see Supplemental Information 1). This new larval preparation is referred to herein as the ‘off-midline dissection’ (Fig. 1).

Figure 1 Off-midline larval preparation.

A cut (dashed line) was made to the right of the midline near muscle 4. Larvae were pinned and cleaned so that the muscles (rectangles) and the central nervous system (solid black) were exposed. Muscles 1, 2, 4, 6, 7 and abdominal segments A5–A7 are labeled. For clarity, not all muscles, segments, or nerves are pictured.

Electrophysiology

Intracellular recordings were made at room temperature (21–23 °C) from dorsal muscles 1 or 2 in abdominal segments 2–6, as described previously by others (Barclay, Atwood & Robertson, 2002; Cattaert & Birman, 2001). In the majority of experiments, muscles in two adjacent segments were simultaneously recorded, while in a few experiments muscle activity from only one segment was recorded. Sharp electrodes were pulled from thin-walled borosilicate glass on a P-87 Flaming/Brown filament puller (Sutter Instrument Co.) to a resistance of 30–50 MΩ. Using a long and flexible tip was crucial for allowing the electrode to move with the muscle during peristaltic waves of contractions. Electrodes were filled with 3 M KCl or KAc for recording. Recordings were made with an Axoclamp 2B amplifier (Molecular Devices) in bridge mode and digitized at a sampling rate of 10 kHz by a Digidata 1320A (Axon Instruments). Data were stored using PClamp 8.2 (Molecular Devices) and imported into Spike2 (Cambridge Electronic Design).

Data analysis

Preparations were observed through an Olympus BX51WI microscope. The incidence of visible peristaltic waves, including the direction of the waves, was noted manually and marked with electronic timestamps to restrict analysis to these bouts. Activity such as tonic firing, or bursts of action potentials not associated with peristaltic waves, was not included in the analysis. The following criteria had to be met for a preparation to be considered rhythmically active: (1) at least 3 spontaneous and consecutive posterior (P) to anterior (A) or A to P waves were recorded, (2) the minimum frequency of the activity was 3 bursts per minute, and (3) the bout was at least 1 min in duration; bursts occurring more than 1 min apart were considered to belong to separate bouts.

Criteria to include rhythmic activity in the analysis of the motor pattern were more stringent. In addition to satisfying (1)–(3), only P to A wave activity was included, since this was the prevalent type of activity. The determination of wave type had to be both visually confirmed, and supported by appropriate segmental delays in the recordings. The only exceptions to this latter condition were the few single channel recordings that were included based only on visual confirmation of the wave type. Finally, irregular bursting activity that could not be distinguished from wave-related activity was not included. These stricter criteria meant that the number of rhythmically active larvae was often larger that the number whose activity were used for quantification of the motor pattern.

Burst start and end times were marked manually in Spike2 by placing cursors at the beginning of the upstroke of the first spike and the beginning of the downstroke of the last spike, respectively. Timestamps were exported as .csv files. Custom code was written in Python version 2.7 to extract burst durations, cycle durations, duty cycles, and quiescence intervals from the preprocessed data. Burst duration was calculated as the time elapsed between the start and end of a burst. Cycle duration was calculated as the time elapsed between start times of successive bursts. Duty cycle was obtained by dividing burst duration by cycle duration. Quiescence interval was calculated as the time elapsed between the end of one burst and the start of the next.

Many previous studies analyzing bursting activity in a population have pooled all observations of a particular measure (e.g. all burst durations), irrespective of the animal in which they were recorded, and performed analyses on these pooled data (e.g. in Drosophila see Fox, Soll & Wu (2006)). This violates the assumption of many statistical tests that the observations in a sample are independent of one another (Hoel, Port & Stone, 1971). Multiple bursts gathered from the same animal are not independent measures, and dealing with them as such constitutes pseudoreplication (Lazic, 2010). Instead, quantiles from each measure were calculated for single animals to approximate the individual probability distribution functions. These approximated distributions were then averaged. The average distribution is representative of an average random variable, which can be compared across groups. Each animal is represented only once in the final analysis, thereby avoiding pseudoreplication. Note that the averaging procedure does not assume a particular distribution of the data. This procedure is described in more detail in McKiernan (2010).

Studies of bursting activity often report the mean and standard deviation to compare data sets. However, such measures are only representative when the data are normally distributed. The data obtained from these experiments were not normally distributed (confirmed by tests for normality; data not shown) and thus were poorly described by a single quantity like the mean. Therefore, for each of the measures, the minimum, maximum, and quartile values are reported to give a more complete description of the distribution. The only exceptions are cycle duration and quiescence interval for which precise maximum values are not reported. This is because all groups included animals whose bursting activity was comprised of bouts separated by 1 min or more. Thus, the maximum cycle duration and quiescence interval were reported as ≥ 60 s. The first quartile (Q1) is the value at or below which 25% of the data fall in the distribution. The second quartile (Q2), also known as the median, splits the distribution in half. The third quartile (Q3) is the value delineating 75% of the distribution. To statistically test differences between groups, the Mann-Whitney U test (also known as the rank sum test) was used due to its lack of assumptions about the distribution of the data and its ability to test for shifts in one distribution relative to another (Hart, 2001; Hoel, Port & Stone, 1971; Mendenhall & Ott, 1980). The threshold for significance was set at 0.05.

Results

slo mutants are rhythmically active

To investigate the effects of altering slo channel expression on rhythmic motor activity, recordings were made from mutant larvae expressing the slo1 allele. Expression of slo1 has been shown to largely reduce or eliminate the BK current in larval (Komatsu et al., 1990; Singh & Wu, 1989; Singh & Wu, 1990) and adult (Elkins, Ganetzky & Wu, 1986) muscles, and neurons (Saito & Wu, 1991). Dual intracellular recordings were made from dorsal muscles 1 (M1) or 2 (M2) in adjacent abdominal segments in 18 slo1 off-midline dissected larvae. Their activity was compared to WT (w1118) larvae prepared with the same dissection.

Rhythmic peristaltic wave activity was recorded in 25 of 27 (93%) WT larvae. Both anterior (A) to posterior (P) and P to A waves were observed, with the latter the more prevalent type of activity. P to A waves were recorded in all 25 larvae (100%), while A to P waves were only recorded in 5 larvae (20%). A total of 24 bouts were recorded across 25 larvae, 20 (83%) of which were comprised of P to A waves, and 4 (17%) of which were combination bouts including both wave types. No bouts of exclusively A to P waves were recorded. Individual P to A wave bouts ranged from 1 to 9 min, while combination bouts lasted 4–9 min.

17 of 18 slo1 larvae (94%) displayed spontaneous bouts of rhythmic peristaltic activity. As in WT larvae, both A to P and P to A waves were observed in mutant larvae, with the latter being more prevalent. Of 16 active larvae in which wave type was determined, 15 of 16 (94%) showed P to A waves, while just 3 of 16 (19%) displayed A to P waves. Of 18 total recorded bouts, 14 (78%) consisted of exclusively P to A waves and 4 (22%) of only A to P waves. P to A waves bouts ranged from 1 to 9 min, while A to P wave bouts were between 2 and 6 min. Thus, the incidence and duration of rhythmic bouts in slo1 larvae were not different from WT (p > 0.05).

slo mutants show faster rhythmic activity than WT

Next, to determine if the timing of the motor pattern was altered due to slo1 expression, P to A wave activity was quantified in 21 of 25 active WT larvae and 10 of 17 active slo1 larvae. According to the more stringent criteria to include activity in the analysis (see Methods), recordings from 7 mutant larvae were excluded. This represents a larger exclusion percentage (41%) than that seen in WT (4 larvae; 16%), and was mostly due to irregular bursting that could not be separated from wave-related activity in those animals (Fig. 2C). In other words, though 94% of slo1 larvae were capable of producing rhythmic activity, this group did show an increased propensity for irregular bursting.

Figure 2 slo1 mutant larvae display altered rhythmic motor output.

A. Dual intracellular recordings from WT (top two traces) and slo1 (bottom two traces) larvae. Activity was recorded during P to A waves in adjacent M1’s of segments A4 and A5, as indicated. Scale bar is 10 s. B. Enlargement of dual recordings in A from WT and slo1 larvae. Scale bar is 2 s. C. Examples of irregular bursting activity not associated with peristaltic waves recorded in two slo1 larvae. Scale bar is 10 s.

Figure 2 includes representative recordings of P to A wave activity from WT and slo1 larvae showing a regular, but much faster, motor pattern in mutant animals. The histograms of burst duration, cycle duration, duty cycle, and quiescence interval were all clearly shifted relative to WT (Fig. 3). Minimum, maximum, and quartile values for each measure from WT and slo1 larvae are presented for comparison in Table 1. Differences between the quartile values revealed that burst durations in mutant larvae were 2.16–2.55 seconds (s) shorter (25–40% decrease) relative to WT, while cycle durations were shorter by 4.48–4.84 s (35–45% decrease). The greater decrease in cycle duration relative to burst duration resulted in duty cycles that were larger by 0.06–0.07 (8–9% increase) in mutant larvae. Quiescence intervals were shorter by 2.03–2.78 s (54–56% decrease). In sum, expression of the slo1 mutant allele decreased burst duration, cycle duration, and quiescence interval, and increased duty cycle, relative to WT (p < 0.001 on all comparisons).

Table 1 Measures of motor pattern in WT, slo1, and slo RNAi larvae.

Genotype	Measure	n	min (s)	max (s)	Q1 (s)	Q2 (s)	Q3 (s)	p-valuea	p-valueb	
WT	Burst duration	21	1.61	17.21	6.32	7.19	8.44	-	-	
Cycle duration	21	5.19	≥60	10.66	11.81	12.93	-	-	
Duty cycle	21	0.05	0.93	0.57	0.63	0.69	-	-	
Quiescence interval	21	0.59	≥60	3.63	4.35	5.18	-	-	
slo 1	Burst duration	10	1.15	27.35	3.77	4.78	6.28	<0.001	-	
Cycle duration	10	1.89	≥60	5.81	6.94	8.45	<0.001	-	
Duty cycle	10	0.03	0.96	0.63	0.69	0.75	<0.001	-	
Quiescence interval	10	0.26	≥60	1.60	1.92	2.39	<0.001	-	
slo RNAi	Burst duration	22	2.35	29.72	6.44	7.73	9.41	>0.05	>0.05	
Cycle duration	22	4.91	≥60	9.31	10.62	12.55	<0.001	<0.001	
Duty cycle	22	0.08	0.98	0.67	0.73	0.79	<0.001	<0.01	
Quiescence interval	22	0.33	≥60	2.09	2.74	3.60	<0.001	<0.001	
Notes.

a Mann-Whitney U Test/Wilcoxon rank-sum test, compared to WT.

b Mann-Whitney U Test/Wilcoxon rank-sum test, compared to slo1.

Figure 3 Quantification of motor activity in slo1 larvae.

Histograms of burst durations (A), cycle durations (B), duty cycles (C), and quiescence intervals (D) for WT (black) and slo1 (white) larvae.

Rhythmic activity in slo1 larvae was quantitatively different on all measures from that of WT, demonstrating that altering slo expression in Drosophila larvae significantly alters the timing of the locomotor pattern. However, because the mutant allele was expressed in every cell of the animal, it was not possible from these results to determine in which group(s) of cells the loss of slo channels exerted its effects on the motor pattern. To start addressing specificity, and in particular to explore the influence of MNs on the motor pattern, the next step was to determine whether the observed effects in the mutant could be phenocopied by restricting manipulation of slo expression to identified MNs.

slo RNAi expression in MNs alters cycle, but not burst, duration

To explore the effects of altering slo expression in MNs, the UAS-GAL4 system (Brand & Perrimon, 1993) was used to target a slo RNAi construct to identified MNs. This UAS-slo RNAi construct has been used previously by others (Kwon et al., 2010; Lee & Wu, 2010), and its pan-neuronal expression results in a ∼30% decrease in slo mRNA (Scheckel, 2011). Dicer was added to the driver construct to further increase the strength of the knockdown (Dietzl et al., 2007). Expression of slo RNAi was targeted to two identified MNs per nervous system hemisegment, MN1-Ib and MNISN-Is (Hoang & Chiba, 2001), using the RRA-GAL4 driver (Fujioka et al., 2003). Dual intracellular recordings were made from the target muscles of these two MNs, M1 or M2 (Hoang & Chiba, 2001), in RRA-GAL4; UAS-slo RNAi (abbreviated slo RNAi) larvae. Their activity was compared to WT larvae.

22 of 23 slo RNAi larvae (96%) displayed spontaneous bouts of rhythmic peristaltic activity. As in WT larvae, both A to P and P to A waves were observed in these larvae, with the latter the more frequently recorded activity type. All active larvae showed P to A waves, while only 3 (14%) displayed A to P waves. Of 25 total bouts in which wave type was confirmed, 22 (88%) consisted of exclusively P to A waves and 3 (12%) were combination bouts consisting of both wave types. No bouts comprised of only A to P waves were recorded in slo RNAi larvae. P to A waves bouts ranged from 1 to 10 min, while combination bouts were between 6 and 9 min. Thus, expression of slo RNAi in identified MNs did not alter the incidence or duration of rhythmic motor activity compared to WT larvae (p > 0.05).

To determine whether the timing of the motor pattern was altered by slo RNAi expression in MNs, burst duration, cycle duration, duty cycle, and quiescence interval were quantified in all 22 active larvae. Figure 4 shows representative recordings of P to A wave bouts from WT and slo RNAi larvae. The recordings revealed a similar, though slightly faster, motor pattern in slo RNAi animals relative to WT. Shifts in the relative frequencies calculated from WT and slo RNAi recordings were seen for all measures, except burst duration (Fig. 5). Minimum, maximum, and quartile values for each measure from WT and slo RNAi larvae are presented for comparison in Table 1. While there was no difference in burst duration between the two groups (p > 0.05), cycle durations were shorter in slo RNAi compared to WT larvae by 0.38–1.35 s (3–13% decrease). Since burst duration did not change, while cycle duration decreased, duty cycles were larger by 0.09–0.10 (13–15% increase) in RNAi larvae. The largest difference between the groups was with respect to quiescence interval, which was smaller in RNAi larvae by 1.55–1.61 s (30–42% decrease) relative to WT. In fact, in some slo RNAi larvae, a low level of tonic firing was present between bursts and no significant period of quiescence was recorded (Fig. 4C), even though regular peristaltic waves were still visually observed. In sum, expression of slo RNAi in MNs decreased cycle duration and quiescence interval and increased duty cycle, relative to WT (p < 0.001 on all comparisons), thereby phenocopying 3 of the 4 effects on the motor pattern seen in slo1 mutants.

Figure 4 slo RNAi larvae show altered rhythmic motor output.

A. Dual intracellular recordings from WT (top two traces; from same recording as shown in Fig. 2A) and larvae expressing slo RNAi under the control of RRA-GAL4 (bottom two traces) during P to A waves. Scale bar 10 s. B. Enlargements of recordings in A for WT and slo RNAi larvae. Scale bar 2 s. C. Recording from slo RNAi larva showing low-level tonic firing between bursts. Top scale bar 10 s, bottom scale bar 2 s.

To test whether there was a difference between manipulating slo expression in the whole animal versus in select MNs, a statistical comparison was done between slo1 and slo RNAi larvae. Burst durations in slo RNAi larvae were comparable to WT and thus significantly longer than those in slo1 mutants (p < 0.001). Cycle durations and quiescence intervals were larger in slo RNAi larvae than in slo1 mutants (p < 0.001), having decreased to a lesser extent relative to WT. Duty cycles were also slightly larger in slo RNAi compared to slo1 larvae (p < 0.01). Thus, the two manipulations had qualitatively similar, though quantitatively distinct, effects on the motor pattern.

Figure 5 Quantification of motor activity in slo RNAi larvae.

Histograms of burst durations (A), cycle durations (B), duty cycles (C), and quiescence intervals (D) for WT (black) and slo RNAi (white) larvae.

Discussion

As genetic techniques identify an increasing number of ion channel genes and splice variants (Vacher, Mohapatra & Trimmer, 2008; Wicher, Walther & Wicher, 2001), the challenge for neurophysiologists is to understand how each of these contribute to the behavioral output of the nervous system. In particular, increasing attention is being paid to the role that currents encoded by specific genes can play in shaping the firing patterns of MNs in response to synaptic input (Harris-Warrick, 2002; Heckman et al., 2009; Kiehn et al., 2000; Marder & Goaillard, 2006). This work explored the effects of manipulating the expression of slowpoke Ca2+-dependent K+ channels, in the whole organism and in identified MNs, on the production and timing of rhythmic locomotor activity in Drosophila.

slo manipulation does not interfere with production of rhythmic motor output

Although expression of a slo mutation in the whole larva resulted in a slight increase in the incidence of irregular bursting, the majority of slo1 mutants were able to generate sustained bouts of visible peristaltic waves and corresponding rhythmic motor activity. Similarly, expression of slo RNAi in MNs did not interfere with the ability of the system to produce rhythmic motor output, as measured from the MN muscle targets. These results indicate that either slo currents do not contribute significantly to rhythm generation in the larva, or expression of the manipulations throughout development induced compensatory changes in expression of other ion channels.

Previous work at the Drosphila neuromuscular junction has shown that the voltage-gated A-type K+ channel gene, Shaker, is upregulated in slo mutants, helping to correct abnormalities in synaptic transmission caused by the loss of slo channels (Lee, Ueda & Wu, 2008). An increase in Shaker mRNA has also been observed in Drosophila cultured neurons in response to a reduction in IKCa (Peng & Wu, 2007). Thus, the ability of slo1 and slo RNAi larvae to produce rhythmic motor activity could be due in part to upregulated expression of other K+ channels, including but not limited to A-type channels. Similar homeostatic mechanisms that preserve rhythmic motor output in the face of variable current densities, and specifically compensatory upregulation of A-type channels, have been reported in other systems (MacLean et al., 2003; Marder & Goaillard, 2006). Future work to address the issue of homeostatic compensation should include the acute manipulation of slo expression using inducible GAL4 (Osterwalder et al., 2001).

slo manipulation alters the frequency of rhythmic motor activity

Genetic manipulation of slo in both the whole animal and identified MNs significantly altered the timing of the motor pattern. slo1 larvae displayed much faster motor patterns than WT larvae, with substantial decreases in burst duration, cycle duration, and quiescence interval, and an increase in duty cycle. These effects, with the exception of the decrease in burst duration, were phenocopied by expression of slo RNAi in MNs, though the changes were smaller in magnitude than those seen in slo1 larvae. It could be that the larger increase in the frequency of motor activity recorded from slo1 larvae was due to expression of the mutant allele outside MNs. Or, the relatively less severe phenotype of the slo RNAi larvae could be because the slo mutation results in a more significant loss of functional channels than expression of the RNAi construct (Scheckel, 2011). The differences seen between the mutant and RNAi lines could also be due to distinct compensatory mechanisms. While pan-neuronal expression of slo RNAi does not appear to upregulate the expression of K+channels encoded by SK, Shaker, Shal, or eag, slo1 mutants do show an increase in eag mRNA. Characterizing how ion channel compensation differs in mutants and RNAi lines will therefore be important when comparing their effects on motor output.

Overall, the results from both sets of experiments demonstrate that manipulating slo expression alters the frequency of rhythmic motor activity underlying crawling. It remains to be determined whether these effects are due only to the loss of slo channels, or due to other changes in ion channel expression that result from manipulating slo. It should be noted that these results regarding the increased frequency of locomotor activity in slo1 larvae are in contrast to a previous study (Guan et al., 2005). Although they did not fully characterize the motor pattern, Guan et al. (2005) report that the frequency of bursting was reduced in Drosophila slo1 larvae relative to WT when recorded at 21 °C (Guan et al., 2005). The only obvious distinction between their study and this work was the use of the off-midline dissection method. Larvae dissected off the midline show faster motor activity than those dissected on the midline (Supplemental Information 1), suggesting sensory feedback in this system contributes to regulating the frequency of locomotor activity, as in other motor systems (Grillner et al., 1995). However, since the activity of mutant animals was compared only to WT animals prepared using the same method, the dissection should not have affected the results. At this time, the reason for the discrepancy is unknown, but it is interesting to note that similar conflicting results have been reported in studies of C. elegans slo mutants. While some studies have reported that slo mutants display similar or slightly slower rates of locomotion than WT (Wang et al., 2001), others have reported that these mutants move faster (Carre-Pierrat et al., 2006). Even those studies reporting normal rates of locomotion have found that expression of slo mutant alleles can rescue other mutations with inhibitory effects on locomotion (for review see Holden-Dye et al., 2007).

Possible mechanisms for regulation of bursting frequency by slo currents

How might slo currents shape the frequency of rhythmic motor activity and, in particular, how could cycle duration be shortened by expression of slo RNAi in MNs? There are at least two possible explanations. The first relates to the specificity of the driver, RRA-GAL4, used to target RNAi expression to identified MNs. In addition to MN1-Ib and MNISN-Is, RRA-GAL4 expresses in an interneuron known by its embryonic identity as the posterior corner cell, pCC (Fujioka et al., 2003). Therefore, in the experiments described herein slo RNAi was likely also expressed in pCC. If pCC provides input to MNs, either directly or indirectly, then a change in the activity of this interneuron could explain a change in cycle duration. Future experiments to determine the role of pCC in producing the locomotor phenotype observed in slo RNAi larvae could include the mosaic expression of slo RNAi using the flippase/FLP recognition target (FLP/FRT) system (Golic & Lindquist, 1989). Larvae could be screened and only those without pCC expression recorded to see whether the decrease in cycle duration is still observed. If such studies support a role for pCC in regulating the frequency of MN bursting, it would be the first identification, to the author’s knowledge, of a specific interneuron contributing to generation of crawling rhythms in Drosophila. Identification of such a neuron could pave the way for studies of motor circuitry using the advanced genetic tools for which this model system is renowned.

On the other hand, it is possible that expression of slo RNAi in MNs, and not pCC, was responsible for the decrease in cycle duration. The role of MN currents in shaping rhythmic motor activity is a subject of active research (Harris-Warrick, 2002; Heckman et al., 2009; Kiehn et al., 2000; Marder & Goaillard, 2006). Much of the recent work has focused on the role of persistent inward currents in MNs (Enríquez Denton et al., 2012; Manuel et al., 2012; Revill & Fuglevand, 2011), but outward currents have also been shown to shape MN firing during rhythmic activity (Hooper et al., 2009; Manuel et al., 2012; Ping et al., 2011; Schaefer, Worrell & Levine, 2010). In particular, a recent study in Drosophila showed that elimination of Shal-mediated currents in MNs, via expression of a dominant negative construct under the control of RRA-GAL4, altered the frequency of larval crawling (Ping et al., 2011). The results from the RNAi experiments reported herein add to an increasing body of work suggesting that MN outward currents, and BK currents in particular, may enable MNs to shape the timing of rhythmic motor output. Specifically, the large decrease in quiescence interval in slo RNAi larvae suggests that slo currents may act in MNs as a hyperpolarizing force following a burst (Womack & Khodakhah, 2004). Such a force would counteract inward currents to keep MNs from firing, contributing to the creation of a period of quiescence. Reduction or elimination of slo currents would shorten the quiescence interval and thereby decrease the cycle duration, as observed in this study.

Previous studies have reported that blocking BK channels decreases quiescence interval and burst duration (Wang, Olshausen & Chalupa, 1999; Womack & Khodakhah, 2004). Womack & Khodakhah (2004) hypothesized that BK channels provide a hyperpolarizing influence between bursts, producing quiescence. To explain why blocking BK channels shortened quiescence interval while also decreasing, rather than increasing, burst duration, they speculate that BK channels activate late or after bursting has ceased. Another mechanism, not involving BK channels, terminates the bursts. They go on to explain that the burst-terminating mechanism can activate sooner when BK channels are blocked, causing the decrease in burst duration. Similarly, in Drosophila larvae, slo channels may maximally activate late, after another mechanism has already terminated, or is in the process of terminating, the burst. This could explain why slo RNAi larvae showed no change in burst duration, even though quiescence interval decreased. If in some neurons a reduction in slo channel expression also allows whatever mechanism terminates bursts to do so sooner, this could explain why burst duration decreased in slo1 mutants.

A recent study by Pulver & Griffith (2010) reported that after-hyperpolarizations following bursts in Drosophila larval MNs are mediated by the Na+/K+ ATPase, and not by IKCa (Pulver & Griffith, 2010). However, their results are not necessarily in conflict with the hypothesis that slo currents contribute to setting the quiescence interval. First, it is possible that slo currents may counteract inward currents and keep the cell from firing without producing a measurable after-hyperpolarization. Second, and mentioned by the authors themselves, is the importance of looking at endogenous activity when examining the role of a particular current. Rather than recording spontaneous bursting in MNs, the authors stimulated MNs with trains of square pulses at a frequency designed to mimic rhythmic synaptic input. The activation (and inactivation) of ionic currents could be very different in response to square-pulse stimulation, as opposed to endogenous synaptic input. Intracellular recordings from MNs during spontaneous locomotor activity, as recently described by others (Schaefer, Worrell & Levine, 2010), will be crucial for determining what currents contribute when bursting is driven by endogenous inputs.

Limitations, open questions, and future directions

The biggest limitation of the current work is the lack of a full characterization of the slo RNAi knockdown. Preliminary RT-PCR results indicate that pan-neuronal expression of the UAS-slo RNAi construct under the control of elav-GAL4 produces a ∼30% decrease in slo mRNA when quantified in the whole nervous system (Scheckel, 2011). However, these data do not indicate the extent of the reduction in slo mRNA in the MNs targeted in this study. It is possible manipulation of slo was unsuccessful or insignificant in all or some MNs, though this seems unlikely since in that case one would not expect to observe changes in the motor pattern relative to WT, as were presented.

However, if the manipulation was successful in some and not other animals, this could potentially change the interpretation of the results. For example, the histogram of burst durations in slo RNAi larvae largely overlaps with the WT histogram (Fig. 5), leading to the conclusion that slo manipulation does not affect burst termination in larval MNs. The histogram does include, however, a small tail consisting of a few recordings in which some burst durations were longer than those recorded in WT. If the knockdown was more effective in these animals, this could explain these outlying values and suggest that slo currents may play a role in burst termination. In contrast, the histograms for other measures such as cycle duration and quiescence interval, where it was concluded that the knockdown had an effect, do not show any obvious tails or bimodalities that would indicate distinct efficacies of the knockdown.

To conclude whether there is a relationship between the extent of the knockdown in a particular MN and the recorded motor pattern would require performing one of two possible experiments. First, one could record from the animal and later perform single cell RT-PCR on MNs expressing the knockdown to measure the amount of slo mRNA. A second possibility is that one could record the motor pattern and later do patch clamp recordings from MNs using pharmacology to isolate and measure the slo current. Both options present significant technical challenges, but are potential future directions. A less technically challenging, but also less informative, approach would be to patch on to several MNs to determine the percentage showing a significant reduction in slo current and correlate that with the percentage of animals showing a change in motor pattern.

Additional limitations relate to information about the model system itself. The slo current has not been fully characterized in larval MNs. It remains unclear whether slo channels carry transient and/or sustained Ca2+-dependent K+ currents, and to what extent these currents contribute to the whole-cell current in larval MNs. This information is vital to understanding the effects of slo knockdown on MN excitability. Obtaining this information will require patch clamp recordings using pharmacological blockers specific to the slo current to isolate it from other Ca2+-dependent currents in MNs. In addition, the circuitry of the putative locomotor CPG is unknown in this system. An understanding of the network connectivity, and in particular whether MNs participate in generating the rhythm itself (as in the crustacean STG (Marder & Bucher, 2001)) is crucial for interpreting the observed changes in the frequency of the locomotor rhythm. To map the circuitry of the CPG will be challenging and require a clever combination of genetic techniques and simultaneous recordings from neurons in multiple regions of the larval nervous system.

Conclusions

The current work shows that manipulating the expression of slo channels significantly alters the timing of locomotor activity in Drosophila larvae, and specifically points to the importance of MN currents in shaping motor output. Several questions remain open regarding exactly how slo currents affect MN firing and how changes in MN excitability may interact with synaptic inputs to alter the frequency of motor activity. Alternative hypotheses and possible experiments have been outlined that, if completed, will shed more light on this motor system, and more generally on our understanding of motor control. It is hoped that this work will serve as a basis for future studies on the role of slo channels in locomotor behavior and the investigation of the relative contribution of network versus MN intrinsic properties in the generation of rhythmic motor output.

Supplemental Information

Supplemental Information 1 Effects of dissection method on locomotor activity in Drosophila larvae.

Comparing rhythmic locomotor activity recorded from larvae dissected using the dorsal-midline or off-midline dissection.

Click here for additional data file.

The author thanks Marco A. Herrera Valdez and Peyman Golshani for valuable feedback on earlier drafts. The author is also grateful to Fabiana Kubke, Rune Berg, and one anonymous reviewer, whose suggestions improved this article.

Additional Information and Declarations

Competing Interests

Author Contributions

The author declares that there are no competing interests.

Erin C. McKiernan conceived and designed the experiments, performed the experiments, analyzed the data, contributed reagents/materials/analysis tools, wrote the paper.

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
