# Peer review of "Effects of manipulating slowpoke calcium-dependent potassium channel expression on rhythmic locomotor activity in Drosophila larvae"

_PeerJ, doi:10.7717/peerj.57_

## Round 0.1 · original submission · Minor Revisions

I have receivedtwo reviewer’s reports and have reviewed the manuscript myself. Both the reviewer and myself agree that this is an interesting study that adds value to the field. Some suggestions are made to improve the manuscript that I hope you can address before the manuscript is accepted for submission.

The main issue raised by reviewer #2 is whether the data presented provides sufficient support for the role of BK channels in shaping the locomotor rhythm, a reasoning I tend to agree with, and the issues are more of a semantic nature rather than requiring any new experiments or analysis.

One question that came to mind regarding the involvement of slo in the motor pattern, was how much of the observed effects might be due to the compensatory effects that may accompany the manipulations as opposed to the deficits in slo itself. What might the data look like if one were able to just knock out slo without any other changes in the system? It seems to me that it is hard from this approach to actually isolate the contribution of slo to the motor pattern itself, but rather what one is looking at in these data is what is the effect of the slo manipulation (which is accompanied by a whole sort of other changes). Similarly, given the differences between the results for slo1 and sloRNAi (e.g., burst duration) the question may be also raised about whether one can distinguish between different
compensatory mechanisms being triggered by the two treatments rather than differences in the loss itself of slo1 globally or just in MNs. I think it might be useful to discuss this a bit further.

Methodological queries (mine):

On P3, par 2 you state that data from each animal was averaged – would this assume normality of distribution (as is addressed in par3). I am not sure I have much of a problem with it, but if the data is not normal, it seems to contradict the validity of averaging that is raised a bit below in the text.

On P3, par 3 it reads “If multiple bouts were recorded in the same animal […]”, but paragraph 2 says this would be represented in the single average?

Should a statistical comparison between the results of slo1 and sloRNAi be made to see if there is a difference between affecting slo globally or only in the monotoneurons?

Are there any expression controls to show how slo is affected?
Reviewer #2 also makes several additional suggestions to improve the readability of the article that should be easy to address.
.

Reviewer 1 ·

Basic reporting

I believe the text and the figuares are appropriate.

Experimental design

OK.

Validity of the findings

The findings appear clearcut but limited in scope.

Additional comments

This study deals with the role of calcium dependent K+ channels (BK) and their role for locomotion in Drosophila larvae. The motor pattern is compared between the wild type and mutations in the BK channels. The larvae still generate the motor pattern but at a higher frequency. This thus means that BK contributes to the rhythm generating circuits, but the synaptic interaction in combination with other cellular properties are sufficient to generate the rhythmic pattern. These experiments are complemented with a slo RNA interference construct that confirm the conclusions. The experiments are well presented and the figures appropriate, and the results appear to be clear but somewhat limited in scope.

·

Basic reporting

In this study, McKiernan investigates the rhythmic activity in the Drosophila larva motoneurons of muscles in the body wall that produce the crawling motion. In particular, the role of the BK channel is investigated via interfering with the gene expressing these channels, the slopoke genes, by both whole animal hypomorphic mutation and RNA interference construct.

The main findings are that the hypomorphic mutation cause the peristaltic waves (which is interpreted as locomotion) to increase in frequency (quiescent period is shortened) as well as shortening in the burst duration. The RNAi experiments has less clear cut results (burst duration seem unchanged), but otherwise similar. These findings are interpreted as the slo -channels being important for regulating the locomotor rhythm.

The study takes the classic reductionist approach of attempting to explain a macroscopic phenomenon (locomotion) via the molecular parts (ion-dynamics of a channel in one neuron). The experimental paradigm is clear and the analyses and methods are solid. The manuscript is clear, well written and well-organized.

Minor comments

Text:
page 3, first column 6 lines from bottom: replace "geq" with $\geq$

There are abbreviations in the manuscript, e.g. MN1-1b, MNISN-1s, RRA-GAL4, pCC. Definitions of those terms may be helpful to the general reader.


Display items:
Suggesting figure panels 3 A, B and D have same abscissa (up to 30 sec) for easier comparison. Same for figure 5 (abscissa from 0-40 sec)

Experimental design

No comments

Validity of the findings

No comments

Additional comments

Though the study is interesting with clear important results, I do not find it convincing that the BK channels, on the basis of the data, has the suggested role of shaping the locomotor rhythm. The reader is left with a number of unaddressed issues.

First, it is important to consider how many neurons are involved in the motor circuitry. The author mentions that the pCC interneurons could potentially also be affected by the manipulation - approximately how many neurons are involved in this circuitry? How do we know it is not a network effect? To what extend does the mutation affect other neurons and their emergent network activity. What drives the rhythm? There must be a substantial synaptic component, potentially overriding the intrinsic properties, since there is a AP and PA wave components.

Second, the mechanistic interpretation is incomplete. I can understand that the slo current works as a hyperpolarizing force following the burst that helps create a period of quiescence. Reduction in the this current helped reduce the quiescent period (the increase in burst frequency). However, this explanation does not fit all the data since the burst length is either unaffected or decreases in mutants. If the slo currents are activated during the burst (by increase in intracellular Ca2+ for instance), blocking the slo currents should give longer burst - but the data show shorter bursts.

Third, what exactly is ment by "important for shaping the rhythmic motor activity"? Does this mean that the slo makes the frequency modulation in the bursts more smooth? Or does it mean that the slo- current help appropriately change the burst frequency to fit the given locomotor task? It seems like it would be a very rigid mechanism that would only work in animals that do not need to change the locomotor frequency (as perhaps is the case in Drosophila larvae?). How would this slo channels accommodate a more wide range of locomotor frequencies?

---

## Round 0.2 · accepted · Accept

I am satisfied that you have addressed the issues raised during the review process.